# The Existence of Weak Solutions for the Vorticity Equation Related to the Stratosphere in a Rotating Spherical Coordinate System

Wenlin Zhang [1,2], Michal Fečkan [3,4] 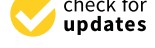 and Jinrong Wang [2,*]

1 School of Mathematics and Statistics, Liupanshui Normal University, Liupanshui 553004, China; wlzhang1025@163.com
2 Department of Mathematics, Guizhou University, Guiyang 550025, China
3 Department of Mathematical Analysis and Numerical Mathematics, Faculty of Mathematics, Physics and Informatics, Comenius University in Bratislava, Mlynská dolina, 842 48 Bratislava, Slovakia; michal.feckan@fmph.uniba.sk
4 Mathematical Institute, Slovak Academy of Sciences, Štefánikova 49, 814 73 Bratislava, Slovakia
* Correspondence: jrwang@gzu.edu.cn

**Abstract:** In this paper, based on the Euler equation and mass conservation equation in spherical coordinates, the ratio of the stratospheric average width to the planetary radius and the ratio of the vertical velocity to the horizontal velocity are selected as parameters under appropriate boundary conditions. We establish the approximate system using these two small parameters. In addition, we consider the time dependence of the system and establish the governing equations describing the atmospheric flow. By introducing a flow function to code the system, a nonlinear vorticity equation describing the planetary flow in the stratosphere is obtained. The governing equations describing the atmospheric flow are transformed into a second-order homogeneous linear ordinary differential equation and a Legendre's differential equation by applying the method of separating variables based on the concepts of spherical harmonic functions and weak solutions. The Gronwall inequality and the Cauchy–Schwartz inequality are applied to priori estimates for the vorticity equation describing the stratospheric planetary flow under the appropriate initial and boundary conditions. The existence and non-uniqueness of weak solutions to the vorticity equation are obtained by using the functional analysis technique.

**Keywords:** stratosphere; vorticity; stream function; weak solution

## 1. Introduction

With the high complexity caused by factors such as the rotation of the Earth and heat input, the observed air flows in the atmosphere can be regarded as a description of large-scale motions (see [1]). For large-scale atmospheric flows, consideration of the full nature of the geometry of the Earth's curved space is essential (see [2]). In order to transcend the limitations of the plane geometry of the $f$-plane approximation, the classical approach is to invoke the weak contribution of the curvature by using the $\beta$-plane approximation (see [3]). However, in contrast to the $f$-plane approximation, the $\beta$-plane approximation does not represent a consistent approximation of the geophysical flow governing equations in the mid-latitude and polar regions (see [4]). Therefore, it is necessary to establish the governing equation of the fluid in the spherical coordinate system.

Constantin and Johnson gave the motion control equation and the mass conservation equation in the rotating spherical coordinate system. The thin-layer asymptotic approximation was established based on the ratio of the ocean average depth to the Earth's radius (see [5]). Martin established the exact solution of the governing equations of geophysical

fluid dynamics involving discontinuous stratification in spherical coordinates (see [6,7]). The general problem of the ocean on a rotating sphere was studied. The exact solution of incompressible (constant density) inviscid fluid with velocity distribution below and along the surface was established. This can be regarded as a model of the Antarctic Circumpolar Current (see [8]). The spherical coordinate governing the equations of inviscid incompressible fluid fixed at a point on the rotating Earth and the free surface and rigid bottom boundary conditions were introduced. The exact solution of the system was given, which described a steady flow that moves only in the azimuth direction (see [9]). Using the fixed point method, Chu established the existence of strictly monotone bounded solutions for a given continuous vorticity. These results were related to the behavior of ocean flows in arctic gyres (see [10]). Wang et al. proposed a non-local formula for simulating the Antarctic Circumpolar Current without considering the vertical motion, which coded the horizontal flow components by introducing flow functions. Using the topological degree, zero exponent theory and fixed point technique, the existence of positive solutions for nonlinear vorticity, nonlocal boundary value problems was proven (see [11]). Haziot used the Mercator projection to map the circulation model from the sphere to the plane and obtained the boundary value problem of semilinear elliptic partial differential equations. For the constant and linear ocean vorticity, he studied the existence, regularity and uniqueness of solutions to this elliptic problem. The physical correlation of these results was also investigated (see [12]). Using spherical coordinates, Martin and Quirchmayr derived a new exact solution to the governing equations of geophysical fluid dynamics for an inviscid and incompressible fluid with a general density distribution and a forced term. Their explicit solution represents a steady purely azimuthal stratified flow with a free surface (see [13,14]). In addition, Constantin and Johnson showed that a consistent shallow-water approximation of the incompressible Navier–Stokes equation written in a spherical, rotating coordinate system produces, at the leading order in a suitable limiting process, a general linear theory for wind-induced ocean currents which reaches beyond the limitations of the classical Ekman spiral (see [15,16]).

In this paper, we consider the spherical coordinates from the Euler equation and mass conservation equation, combined with the appropriate boundary conditions, and choose the ratio of the average width of the stratosphere and planetary radius and the ratio between the vertical velocity and horizontal velocity as parameters. We utilize the two small parameters to establish the approximate system. Furthermore, we consider adding time dependence into the system and establish the governing equations to describe the atmospheric flow. A nonlinear vorticity equation describing stratospheric planetary flow is obtained by introducing a flow function to code the system. Based on the idea of spherical harmonic function and the concept of a weak solution, the existence and non-uniqueness of a weak solution to the vorticity equation are obtained under suitable boundaries.

## 2. Stratospheric Planetary Flows Equation

In this section, we introduce the rotating sphere coordinate system $(\beta, \alpha, \bar{r})$, with $\beta \in [0, 2\pi]$ and $\alpha \in \left[-\frac{\pi}{2}, \frac{\pi}{2}\right]$ representing the longitude and latitude angles, respectively, and $\bar{r}$ representing the distance from the primordial center to the center of the planet. $(\mathbf{e}_\beta, \mathbf{e}_\alpha, \mathbf{e}_{\bar{r}})$ represent the unit vectors of $(\beta, \alpha, \bar{r})$ in the three directions, respectively, and the corresponding velocity components are $(\bar{u}, \bar{v}, \bar{w})$, where $\mathbf{e}_\beta$ points from west to east, $\mathbf{e}_\alpha$ points from south to north, and $\mathbf{e}_{\bar{r}}$ points out from the origin.

In the stratosphere, atmospheric currents can be considered inviscid (see [17]). Under the influence of the Coriolis force, the Euler equation can be determined by the following components (see [18]):

$$\frac{\partial \bar{u}}{\partial \bar{t}} + \frac{\bar{u}}{\bar{r}\cos\alpha}\frac{\partial \bar{u}}{\partial \beta} + \frac{\bar{v}}{\bar{r}}\frac{\partial \bar{u}}{\partial \alpha} + \bar{w}\frac{\partial \bar{u}}{\partial \bar{r}} + \frac{\bar{u}\bar{w} - \bar{u}\bar{v}\tan\alpha}{\bar{r}} - 2\bar{\Omega}(\bar{v}\sin\alpha - \bar{w}\cos\alpha) = -\frac{1}{\bar{\rho}\bar{r}\cos\alpha}\frac{\partial \bar{p}}{\partial \beta},$$

$$\frac{\partial \bar{v}}{\partial \bar{t}} + \frac{\bar{u}}{\bar{r}\cos\alpha}\frac{\partial \bar{v}}{\partial \beta} + \frac{\bar{v}}{\bar{r}}\frac{\partial \bar{v}}{\partial \alpha} + \bar{w}\frac{\partial \bar{v}}{\partial \bar{r}} + \frac{\bar{v}\bar{w} + \bar{u}^2\tan\alpha}{\bar{r}} + 2\bar{\Omega}\bar{u}\sin\alpha + \bar{\Omega}^2\bar{r}\sin\alpha\cos\alpha = -\frac{1}{\bar{\rho}\bar{r}}\frac{\partial \bar{p}}{\partial \alpha}, \tag{1}$$

$$\frac{\partial \bar{w}}{\partial \bar{t}} + \frac{\bar{u}}{\bar{r}\cos\alpha}\frac{\partial \bar{w}}{\partial \beta} + \frac{\bar{v}}{\bar{r}}\frac{\partial \bar{w}}{\partial \alpha} + \bar{w}\frac{\partial \bar{w}}{\partial \bar{r}} - \frac{\bar{u}^2 + \bar{v}^2}{\bar{r}} - 2\bar{\Omega}\bar{u}\cos\alpha - \bar{\Omega}^2\bar{r}\cos^2\alpha = -\frac{1}{\bar{\rho}}\frac{\partial \bar{p}}{\partial \bar{r}} - \bar{g},$$

where $\bar{p}$ and $\bar{\rho}$ represent the atmospheric pressure and density, respectively, $\bar{\Omega}$ is the constant rotation rate of the planet, and $\bar{g}$ stands for the acceleration of gravity.

The mass conservation equation in spherical coordinates can be expressed as

$$\frac{\partial \bar{\rho}}{\partial \bar{t}} + \frac{\bar{u}}{\bar{r}\cos\alpha}\frac{\partial \bar{\rho}}{\partial \beta} + \frac{\bar{v}}{\bar{r}}\frac{\partial \bar{\rho}}{\partial \alpha} + \bar{w}\frac{\partial \bar{\rho}}{\partial \bar{r}} + \bar{\rho}\left[\frac{1}{\bar{r}\cos\alpha}\left(\frac{\partial \bar{u}}{\partial \beta} + \frac{\partial}{\partial \alpha}(\bar{v}\cos\alpha)\right) + \frac{1}{\bar{r}^2}\frac{\partial}{\partial \bar{r}}(\bar{r}^2\bar{w})\right] = 0. \tag{2}$$

The inverse of the Rossby number is expressed as shown below:

$$\mu = \frac{\bar{\Omega}\bar{R}}{\bar{U}},$$

where $\bar{R}$ is the radius of the planet and $\bar{U}$ is the horizontal velocity scale.

The shallowness parameter $\kappa$ and the ratio $\varrho$ between the vertical scale $\bar{W}$ and horizontal velocity scale $\bar{U}$ can be given as follows:

$$\kappa = \frac{\bar{H}}{\bar{R}} \quad \text{and} \quad \varrho = \frac{\bar{W}}{\bar{U}},$$

Here, $\bar{H}$ is the mean width of the stratosphere. Based on the persistent large-scale circulation model of the stratosphere, we can give corresponding reference values for different dimensional scales (see [19–21]). The reference values are shown in Table 1.

We now transform the original variables to receive the dimensionless version of all the variables as follows:

$$(\bar{u}, \bar{v}, \bar{w}) = (\bar{U}u, \bar{U}v, \bar{W}w), \quad \bar{t} = \frac{\bar{R}}{\bar{U}}t, \quad \bar{r} = \bar{R} + \bar{H}z, \quad \bar{\rho} = \tilde{\rho}\rho, \quad \bar{p} = \tilde{\rho}\bar{U}^2 p.$$

Therefore, the dimensionless Euler Equation (1) is transformed into

$$\frac{\partial u}{\partial t} + \frac{u}{(1 + \kappa z)\cos\alpha}\frac{\partial u}{\partial \beta} + \frac{v}{1 + \kappa z}\frac{\partial u}{\partial \alpha} + \frac{\varrho}{\kappa}w\frac{\partial u}{\partial z} + \frac{\varrho uw - uv\tan\alpha}{1 + \kappa z}$$

$$- 2\mu(v\sin\alpha - \varrho w\cos\alpha) = -\frac{1}{\rho(1 + \kappa z)\cos\alpha}\frac{\partial p}{\partial \beta},$$

$$\frac{\partial v}{\partial t} + \frac{u}{(1 + \kappa z)\cos\alpha}\frac{\partial v}{\partial \beta} + \frac{v}{1 + \kappa z}\frac{\partial v}{\partial \alpha} + \frac{\varrho}{\kappa}w\frac{\partial v}{\partial z} + \frac{\varrho vw + u^2\tan\alpha}{1 + \kappa z} \tag{3}$$

$$+ 2\mu u\sin\alpha + \mu^2(1 + \kappa z)\sin\alpha\cos\alpha = -\frac{1}{\rho(1 + \kappa z)}\frac{\partial p}{\partial \alpha},$$

$$\kappa\varrho\left(\frac{\partial w}{\partial t} + \frac{u}{(1 + \kappa z)\cos\alpha}\frac{\partial w}{\partial \beta} + \frac{v}{1 + \kappa z}\frac{\partial w}{\partial \alpha} + \frac{\varrho}{\kappa}w\frac{\partial v}{\partial z}\right) - \kappa\frac{u^2 + v^2}{1 + \kappa z}$$

$$- 2\mu\kappa u\cos\alpha - \kappa\mu^2(1 + \kappa z)^2\cos^2\alpha = -\frac{1}{\rho}\frac{\partial p}{\partial z} - g,$$

where $g = \frac{\bar{g}\bar{H}}{\bar{U}^2}$, and the dimensionless mass conservation Equation (2) can be rewritten as

$$\frac{\partial \rho}{\partial t} + \frac{u}{(1 + \kappa z)\cos \alpha}\frac{\partial \rho}{\partial \beta} + \frac{v}{1 + \kappa z}\frac{\partial \rho}{\partial \alpha} + \frac{\varrho}{\kappa}w\frac{\partial \rho}{\partial z} + \rho\left(\frac{1}{(1 + \kappa z)\cos \alpha}\frac{\partial u}{\partial \beta}\right.$$
$$\left. + \frac{1}{(1 + \kappa z)\cos \alpha}\frac{\partial}{\partial \alpha}(v \cos \alpha) + \frac{\varrho}{\kappa}\frac{1}{(1 + \kappa z)^2}\frac{\partial}{\partial z}\left((1 + \kappa z)^2 w\right)\right) = 0.$$

**Table 1.** Reference values of parameters corresponding to some planets.

| Planet | $\bar{H}$ | $\bar{R}$ | $\bar{g}$ | $\bar{\Omega}$ | $\bar{W}$ | $\bar{U}$ | $\mu$ | $\kappa$ | $\varrho$ |
|--------|-----------|-----------|-----------|----------------|-----------|-----------|-------|----------|-----------|
| Earth | 40 km | 6371 km | 9.8 m/s$^2$ | $7.27 \times 10^{-5}$ rad/s | $10^{-3}$ m/s | 50 m/s | 9 | $6 \times 10^{-3}$ | $2 \times 10^{-5}$ |
| Saturn | 200 km | 58,232 km | 10.4 m/s$^2$ | $1.62 \times 10^{-4}$ rad/s | $10^{-2}$ m/s | 150 m/s | 63 | $3 \times 10^{-3}$ | $6 \times 10^{-5}$ |
| Uranus | 150 km | 25,362 km | 8.8 m/s$^2$ | $1.04 \times 10^{-4}$ rad/s | $10^{-5}$ m/s | 150 m/s | 18 | $6 \times 10^{-3}$ | $6 \times 10^{-8}$ |
| Jupiter | 270 km | 69,911 km | 24.8 m/s$^2$ | $1.76 \times 10^{-4}$ rad/s | $10^{-2}$ m/s | 150 m/s | 82 | $4 \times 10^{-3}$ | $6 \times 10^{-5}$ |
| Neptune | 200 km | 24,622 km | 11.1 m/s$^2$ | $1.08 \times 10^{-4}$ rad/s | $10^{-3}$ m/s | 200 m/s | 13 | $8 \times 10^{-3}$ | $5 \times 10^{-6}$ |

According to the main characteristics of the physical correlation state of the thin shell stratosphere (see [19]), let the shallowness parameter $\kappa \to 0$. Then, the governing Equation (3) is reduziertreduced as

$$\frac{\partial u}{\partial t} + \frac{u}{\cos \alpha}\frac{\partial u}{\partial \beta} + v\frac{\partial u}{\partial \alpha} - uv \tan \alpha - 2\mu v \sin \alpha = -\frac{1}{\rho \cos \alpha}\frac{\partial p}{\partial \beta},$$
$$\frac{\partial v}{\partial t} + \frac{u}{\cos \alpha}\frac{\partial v}{\partial \beta} + v\frac{\partial v}{\partial \alpha} + u^2 \tan \alpha + 2\mu u \sin \alpha + \mu^2 \sin \alpha \cos \alpha = -\frac{1}{\rho}\frac{\partial p}{\partial \alpha}, \qquad (4)$$
$$0 = -\frac{1}{\rho}\frac{\partial p}{\partial z} - g,$$

and

$$\frac{\partial u}{\partial \beta} + \frac{\partial}{\partial \alpha}(v \cos \alpha) = 0. \qquad (5)$$

In Equation (5), the stream function is introduced as follows:

$$u = -\frac{\partial \Psi}{\partial \alpha} \quad \text{and} \quad v = \frac{1}{\cos \alpha}\frac{\partial \Psi}{\partial \beta}. \qquad (6)$$

Substituting Equation (6) into Equation (4) eliminates the pressure term $p$, and by a straightforward calculation, the governing equation of the flow pattern of the atmosphere in the rotating spherical coordinate system can be briefly expressed as

$$\frac{\partial \Delta \Psi}{\partial t} + \frac{1}{\cos \alpha}\left(\frac{\partial \Psi}{\partial \beta}\frac{\partial \Delta \Psi}{\partial \alpha} - \frac{\partial \Psi}{\partial \alpha}\frac{\partial \Delta \Psi}{\partial \beta}\right) + 2\mu\frac{\partial \Psi}{\partial \beta} = 0, \qquad (7)$$

where $\Delta$ is called the Laplace–Beltrami operator on the surface of the unit sphere such that

$$\Delta = \frac{1}{\cos \alpha}\frac{\partial}{\partial \alpha}\left(\cos \alpha\frac{\partial}{\partial \alpha}\right) + \frac{1}{\cos^2 \alpha}\frac{\partial^2}{\partial \beta^2} = \frac{\partial^2}{\partial \alpha^2} - \tan \alpha\frac{\partial}{\partial \alpha} + \frac{1}{\cos^2 \alpha}\frac{\partial^2}{\partial \beta^2}.$$

Equation (7) is called the stratospheric planetary flows equation or simply the vorticity equation in [19]. In [19], the rigidity result of Equation (7) is established, and the Arnold's stability criterion is given under the condition that $\Delta \Psi$ meets the appropriate conditions. The stability of the critical stationary solution is studied in the limit case where the stream function belongs to the sum of the first two eigenspaces of the Laplace–Beltrami operator.

The local and global bifurcation results for the nonzonal stationary solutions of classical Rossby–Haurwitz waves are also obtained. However, it is not difficult to notice that there are no relevant results in the discussion on the existence of weak solutions for general vorticity equations. In this paper, we try to solve this problem.

It is worth mentioning that if the factor of time in Equation (7) is not taken into account, the governing Equation (7) can be regarded as a mathematical model of ocean circulation in which the vertical velocity is relatively weak compared with the horizontal velocity. This was originally developed by Constantin in [5]. After that, many scholars applied functional analysis technology and differential equation theory and found a lot of meaningful results (see [22–33]).

## 3. Main Results

Consider Equation (7) with the associated initial condition

$$\Psi(\alpha, \beta, 0) = \Psi_0(\alpha, \beta), \tag{8}$$

and boundary conditions

$$\Psi(-\pi/2, \beta, t) \text{ and } \Psi(\pi/2, \beta, t) \text{ is bounded },$$
$$\Psi(\alpha, 0, t) = \Psi(\alpha, 2\pi, t). \tag{9}$$

We give the following assumptions:

**Assumption 1.** *Let* $\Psi_0(\alpha, \beta)$, $\frac{\partial \Psi_0}{\partial \alpha}$, $\frac{1}{\cos \alpha} \frac{\partial \Psi_0}{\partial \beta}$, $\Delta \Psi_0 \in L^2\left((-\frac{\pi}{2}, \frac{\pi}{2}) \times (0, 2\pi)\right)$, *further, let us assume that*

$$\Psi_{0n} \to \Psi_0 \quad \text{strongly in} \quad L^2\left((-\frac{\pi}{2}, \frac{\pi}{2}) \times (0, 2\pi)\right) \quad \text{as} \quad n \to \infty,$$

$$\frac{\partial \Psi_{0n}}{\partial \alpha} \to \frac{\partial \Psi_0}{\partial \alpha} \quad \text{strongly in} \quad L^2\left((-\frac{\pi}{2}, \frac{\pi}{2}) \times (0, 2\pi)\right) \quad \text{as} \quad n \to \infty,$$

$$\frac{1}{\cos \alpha} \frac{\partial \Psi_{0n}}{\partial \beta} \to \frac{1}{\cos \alpha} \frac{\partial \Psi_0}{\partial \beta} \quad \text{strongly in} \quad L^2\left((-\frac{\pi}{2}, \frac{\pi}{2}) \times (0, 2\pi)\right) \quad \text{as} \quad n \to \infty,$$

$$\Delta \Psi_{0n} \to \Delta \Psi_0 \quad \text{strongly in} \quad L^2\left((-\frac{\pi}{2}, \frac{\pi}{2}) \times (0, 2\pi)\right) \quad \text{as} \quad n \to \infty,$$

*where* $\Psi_{0n}$, $\frac{\partial \Psi_{0n}}{\partial \alpha}$, $\frac{1}{\cos \alpha} \frac{\partial \Psi_{0n}}{\partial \beta}$ *and* $\Delta \Psi_{0n}$ *are respectively approximate functions of* $\Psi_0$, $\frac{\partial \Psi_0}{\partial \alpha}$, $\frac{1}{\cos \alpha} \frac{\partial \Psi_0}{\partial \beta}$ *and* $\Delta \Psi_0$ *in* $L^2\left((-\frac{\pi}{2}, \frac{\pi}{2}) \times (0, 2\pi)\right)$.

We define the norm as follows:

$$\|\psi\|^2 = \int_0^{2\pi} \int_{-\frac{\pi}{2}}^{\frac{\pi}{2}} \psi^2 \cos \alpha \, d\alpha \, d\beta \quad \text{for} \quad \psi \in L^2\left((0, 2\pi) \times (-\frac{\pi}{2}, \frac{\pi}{2})\right).$$

Now, we turn our consideration to the following boundary value problem:

$$\begin{cases} \Delta \Psi + \lambda \Psi = 0, \\ \Psi(\alpha, 0, t) = \Psi(\alpha, 2\pi, t), \\ \Psi(-\pi/2, \beta, t) \text{ and } \Psi(\pi/2, \beta, t) \text{ is bounded.} \end{cases} \tag{10}$$

By applying the separation of variables technique, we set

$$\Psi = \Theta(\alpha) \cdot \Phi(\beta), \tag{11}$$

By substituting Equation (11) into Equation (10), we have

$$\cos^2 \alpha \frac{\Theta''}{\Theta} - \sin \alpha \cos \alpha \frac{\Theta'}{\Theta} + \lambda \cos^2 \alpha = -\frac{\Phi''}{\Phi} = m^2, \tag{12}$$

where $m$ is a non-negative integer. Equation (12) can be divided into the following two differential equations:

$$\Phi'' + m^2\Phi = 0,\tag{13}$$

and

$$\Theta'' - \frac{\sin\alpha}{\cos\alpha}\Theta' + \left(\lambda - \frac{m^2}{\cos^2\alpha}\Theta\right) = 0.\tag{14}$$

Equation (13) is a second-order homogeneous linear differential equation with constant coefficients, and the general solution of Equation (13) can be expressed as

$$\Phi = \overline{C}_m\cos m\beta + \widetilde{C}_m\sin m\beta.$$

Let $x = \sin\alpha$, $\alpha \in (-\frac{\pi}{2}, \frac{\pi}{2})$, and $\Theta(\alpha) = y(x)$. Then, Equation (14) can be equivalently converted to

$$(1 - x^2)y''(x) - 2xy'(x) + \left(\lambda - \frac{m^2}{1 - x^2}\right)y(x) = 0,\tag{15}$$

Equation (15) is Legendre's differential equation if $m = 0$, and its solution can be expressed as

$$P_l(\sin\alpha) = \frac{1}{\pi}\int_{-\frac{\pi}{2}}^{\frac{\pi}{2}}(\sin\alpha + i\cos\alpha\cos\hat{\alpha})^l d\hat{\alpha}$$

$$= \frac{1}{2^l}\sum_{\iota=0}^{L}(-1)^\iota\frac{(2l - 2\iota)!}{\iota!(l - \iota)!(l - 2\iota)!}(\sin\alpha)^{l - 2\iota},$$

where $l$ is a non-negative integer, $\lambda = l(l + 1)$ is the proper value, and

$$L = \begin{cases} \frac{l}{2}, & l \text{ is even,} \\ \frac{l-1}{2}, & l \text{ is odd.} \end{cases}$$

Let $P_l^m(\sin\alpha)$ be the adjoint function of the Legendre polynomial $P_l(\sin\alpha)$. Then, we have

$$P_l^m(\sin\alpha) = \frac{(l + 1)(l + 2)\cdots(l + m)}{\pi}\int_{-\frac{\pi}{2}}^{\frac{\pi}{2}}(\sin\alpha + i\cos\alpha\cos\hat{\alpha})^l\sin m\hat{\alpha}d\hat{\alpha}.$$

By simple calculation, we can find

$$P_l(\sin\alpha)|_{\alpha=-\frac{\pi}{2}} = (-1)^l, \quad P_l(\sin\alpha)|_{\alpha=\frac{\pi}{2}} = 1,$$
$$P_l^m(\sin\alpha)|_{\alpha=-\frac{\pi}{2}} = 0, \quad P_l^m(\sin\alpha)|_{\alpha=\frac{\pi}{2}} = 0.$$

Let the approximate solution of Equation (7) with the initial condition in Equation (8) and boundary conditions in Equation (9) be

$$\Psi_n(\alpha, \beta, t) = \sum_{l=0}^{n}\sum_{m=0}^{l}\left(\overline{C}_{lm}^n(t)\cos m\beta + \widetilde{C}_{lm}^n(t)\sin m\beta\right)P_l^m(\sin\alpha),\tag{16}$$

By taking the partial derivative, we have

$$\frac{\partial \Psi_n}{\partial \beta} = \sum_{l=1}^{n} \sum_{m=1}^{l} \left[ m \left( -\overline{C}_{lm}^n(t) \sin m\beta + \widetilde{C}_{lm}^n(t) \cos m\beta \right) \right] P_l^m(\sin \alpha),$$

$$\Delta \Psi_n = -\sum_{l=1}^{n} (l(l+1)) \sum_{m=0}^{l} \left( \overline{C}_{lm}^n(t) \cos m\beta + \widetilde{C}_{lm}^n(t) \sin m\beta \right) P_l^m(\sin \alpha),$$

$$\frac{\partial \Delta \Psi_n}{\partial \beta} = -\sum_{l=1}^{n} (l(l+1)) \sum_{m=0}^{l} \left[ m \left( -\overline{C}_{lm}^n(t) \sin m\beta + \widetilde{C}_{lm}^n(t) \cos m\beta \right) \right] P_l^m(\sin \alpha).$$

Suppose that the approximate solution $\Psi_n(\alpha, \beta, t)$ satisfies the following two equations:

$$\frac{1}{\Pi_{lm}} \int_0^{2\pi} \int_{-\frac{\pi}{2}}^{\frac{\pi}{2}} \left[ \frac{\partial \Delta \Psi_n}{\partial t} + \frac{1}{\cos \alpha} \left( \frac{\partial \Psi_n}{\partial \beta} \frac{\partial \Delta \Psi_n}{\partial \alpha} - \frac{\partial \Psi_n}{\partial \alpha} \frac{\partial \Delta \Psi_n}{\partial \beta} \right) + \right.$$
$$\left. 2\mu \frac{\partial \Psi_n}{\partial \beta} \right] \cos m\beta P_l^m(\sin \alpha) \cos \alpha d\alpha d\beta = 0, \tag{17}$$

and

$$\frac{1}{\Pi_{lm}} \int_0^{2\pi} \int_{-\frac{\pi}{2}}^{\frac{\pi}{2}} \left[ \frac{\partial \Delta \Psi_n}{\partial t} + \frac{1}{\cos \alpha} \left( \frac{\partial \Psi_n}{\partial \beta} \frac{\partial \Delta \Psi_n}{\partial \alpha} - \frac{\partial \Psi_n}{\partial \alpha} \frac{\partial \Delta \Psi_n}{\partial \beta} \right) + \right.$$
$$\left. 2\mu \frac{\partial \Psi_n}{\partial \beta} \right] \sin m\beta P_l^m(\sin \alpha) \cos \alpha d\alpha d\beta = 0, \tag{18}$$

where

$$\Pi_{lm} = \int_0^{2\pi} \int_{-\frac{\pi}{2}}^{\frac{\pi}{2}} (P_l^m(\sin \alpha))^2 \cos \alpha \sin^2 m\beta d\alpha d\beta$$
$$= \frac{2\pi \delta_m (l+m)!}{(2l+1)(l-m)!}$$

and

$$\delta_m = \begin{cases} 2, & m = 0, \\ 1, & m \neq 0. \end{cases}$$

By substituting Equation (16) into Equations (17) and (18), the following ordinary differential equations can be obtained:

$$\begin{cases} [-l(l+1)] \frac{d\overline{C}_{lm}^n}{dt} + \overline{F}_{lm}(\overline{C}_{ki}^n, \widetilde{C}_{qj}^n) = 0, \\ [-l(l+1)] \frac{d\widetilde{C}_{lm}^n}{dt} + \widetilde{F}_{lm}(\overline{C}_{ki}^n, \widetilde{C}_{qj}^n) = 0. \end{cases} \tag{19}$$

**Remark 1.** *$\overline{F}_{lm}$ and $\widetilde{F}_{lm}$ are analytic functions of $\overline{C}_{ki}^n$ and $\widetilde{C}_{qj}^n$ ($i = 0, 1, \cdots, k; k = 0, 1, \cdots, n; j = 0, 1, \cdots, q; q = 0, 1, \cdots, n$). When $\overline{C}_{ki}^n$ and $\widetilde{C}_{qj}^n$ are uniformly bounded for any $t \in [0, T]$, $\overline{F}_{lm}$ and $\widetilde{F}_{lm}$ are also bounded, and the Lipschitz conditions are satisfied. According to ordinary differential equation theory, solutions $\overline{C}_{lm}^n(t)$ and $\widetilde{C}_{lm}^n(t)$ of Equation (19) with respect to $t \in [0, T]$ are unique, which also shows that $\frac{d\overline{C}_{lm}^n}{dt}$ and $\frac{d\widetilde{C}_{lm}^n}{dt}$ are uniformly bounded. Therefore, $\overline{C}_{lm}^n(t)$ and $\widetilde{C}_{lm}^n(t)$ are equicontinuous and uniformly bounded with respect to $t \in [0, T]$. By applying the Arzela–Ascoli theorem, $\overline{C}_{lm}^n(\cdot)$ and $\widetilde{C}_{lm}^n(\cdot)$ are compact with respect to $n$.*

**Lemma 1.** *There is a value $M_0 > 0$ such that*

$$\left\|\frac{\partial \Psi_n}{\partial \beta}\right\|^2 \leq M_0 \|\Delta \Psi_n\|^2.$$

**Proof.** On the one hand, we have

$$
\begin{aligned}
\left\|\frac{\partial \Psi_n}{\partial \beta}\right\|^2 &= \int_0^{2\pi} \int_{-\frac{\pi}{2}}^{\frac{\pi}{2}} \left(\frac{\partial \Psi_n}{\partial \beta}\right)^2 \cos \alpha \, d\alpha \, d\beta \\
&= \left| \int_{-\frac{\pi}{2}}^{\frac{\pi}{2}} \int_0^{2\pi} \left[\frac{\partial}{\partial \beta}\left(\Psi_n \frac{\partial \Psi_n}{\partial \beta}\right)\right] \cos \alpha \, d\beta \, d\alpha - \int_0^{2\pi} \int_{-\frac{\pi}{2}}^{\frac{\pi}{2}} \left(\Psi_n \frac{\partial^2 \Psi_n}{\partial \beta^2}\right) \cos \alpha \, d\alpha \, d\beta \right| \\
&\leq \frac{1}{2}\left(\int_0^{2\pi} \int_{-\frac{\pi}{2}}^{\frac{\pi}{2}} \left(\frac{1}{\cos \alpha} \frac{\partial^2 \Psi_n}{\partial \beta^2}\right)^2 \cos \alpha \, d\alpha \, d\beta + \int_0^{2\pi} \int_{-\frac{\pi}{2}}^{\frac{\pi}{2}} (\cos \alpha \Psi_n)^2 \cos \alpha \, d\alpha \, d\beta\right) \\
&\leq \frac{1}{2}\left(\left\|\frac{1}{\cos \alpha} \frac{\partial^2 \Psi_n}{\partial \beta^2}\right\|^2 + \|\Psi_n\|^2\right).
\end{aligned}
$$

On the other hand, we have

$$
\begin{aligned}
\|\Delta \Psi_n\|^2 &= \int_0^{2\pi} \int_{-\frac{\pi}{2}}^{\frac{\pi}{2}} (\Delta \Psi_n)^2 \cos \alpha \, d\alpha \, d\beta \\
&= \sum_{l=0}^{n} \sum_{m=0}^{l} (l(l+1))^2 \left((\overline{C}_{lm}^n)^2 + (\widetilde{C}_{lm}^n)^2\right) \Pi_{lm} \\
&\geq \lambda_0^2 \sum_{l=0}^{n} \sum_{m=0}^{l} \left((\overline{C}_{lm}^n)^2 + (\widetilde{C}_{lm}^n)^2\right) \Pi_{lm} \\
&= \lambda_0^2 \int_0^{2\pi} \int_{-\frac{\pi}{2}}^{\frac{\pi}{2}} \Psi_n^2 \cos \alpha \, d\alpha \, d\beta \\
&= \lambda_0^2 \|\Psi_n\|^2,
\end{aligned}
$$

where $\lambda_0$ is the smallest positive proper value. Furthermore, we have

$$
\begin{aligned}
\|\Delta \Psi_n \cos \alpha\|^2 &= \int_0^{2\pi} \int_{-\frac{\pi}{2}}^{\frac{\pi}{2}} (\cos \alpha \Delta \Psi_n)^2 \cos \alpha \, d\alpha \, d\beta \\
&= \int_0^{2\pi} \int_{-\frac{\pi}{2}}^{\frac{\pi}{2}} \left[\left(\frac{\partial}{\partial \alpha}\left(\cos \alpha \frac{\partial \Psi_n}{\partial \alpha}\right)\right)^2 + \frac{1}{\cos^2 \alpha}\left(\frac{\partial^2 \Psi_n}{\partial \beta^2}\right)^2 \right. \\
&\quad \left. + \frac{2}{\cos \alpha} \frac{\partial}{\partial \alpha}\left(\cos \alpha \frac{\partial \Psi_n}{\partial \alpha}\right) \frac{\partial^2 \Psi_n}{\partial \beta^2}\right] \cos \alpha \, d\alpha \, d\beta \\
&= \int_0^{2\pi} \int_{-\frac{\pi}{2}}^{\frac{\pi}{2}} \left[\left(\frac{\partial}{\partial \alpha}\left(\cos \alpha \frac{\partial \Psi_n}{\partial \alpha}\right)\right)^2\right] \cos \alpha \, d\alpha \, d\beta \\
&\quad + \int_0^{2\pi} \int_{-\frac{\pi}{2}}^{\frac{\pi}{2}} \left[\left(\frac{1}{\cos \alpha} \frac{\partial^2 \Psi_n}{\partial \beta^2}\right)^2\right] \cos \alpha \, d\alpha \, d\beta \\
&\quad + 2 \int_0^{2\pi} \int_{-\frac{\pi}{2}}^{\frac{\pi}{2}} \left[\left(\frac{\partial^2 \Psi_n}{\partial \alpha \partial \beta}\right)^2\right] \cos \alpha \, d\alpha \, d\beta \\
&= \left\|\frac{\partial}{\partial \alpha}\left(\cos \alpha \frac{\partial \Psi_n}{\partial \alpha}\right)\right\|^2 + \left\|\frac{1}{\cos \alpha} \frac{\partial^2 \Psi_n}{\partial \beta^2}\right\|^2 + 2\left\|\frac{\partial^2 \Psi_n}{\partial \alpha \partial \beta}\right\|^2 \\
&\leq \|\Delta \Psi_n\|^2.
\end{aligned}
\tag{20}
$$

Hence, we find

$$\left\|\frac{\partial \Psi_n}{\partial \beta}\right\|^2 \leq \frac{1}{2}\left(\|\Delta \Psi_n\|^2 + \|\Psi_n\|^2\right) \leq \frac{1}{2}\left(\|\Delta \Psi_n\|^2 + \frac{1}{\lambda_0^2}\|\Delta \Psi_n\|^2\right) \leq M_0 \|\Delta \Psi_n\|^2.$$

This completes the proof. $\square$

By multiplying both ends of Equations (17) and (18) by $-l(l+1)\overline{C}_{lm}^{n}(t)$ and $-l(l+1)\widetilde{C}_{lm}^{n}(t)$, respectively, and adding them in turn, we have

$$\int_0^{2\pi}\int_{-\frac{\pi}{2}}^{\frac{\pi}{2}}\left[\frac{\partial\Delta\Psi_n}{\partial t}+\frac{1}{\cos\alpha}\left(\frac{\partial\Psi_n}{\partial\beta}\frac{\partial\Delta\Psi_n}{\partial\alpha}-\frac{\partial\Psi_n}{\partial\alpha}\frac{\partial\Delta\Psi_n}{\partial\beta}\right)+2\mu\frac{\partial\Psi_n}{\partial\beta}\right]\Delta\Psi_n\cos\alpha d\alpha d\beta=0.$$

By using the Cauchy–Schwartz inequality and the properties of the derivatives, we obtain

$$\frac{1}{2}\frac{\partial}{\partial t}\left\|\Delta\Psi_n\right\|^2+\frac{1}{2}\int_0^{2\pi}\int_{-\frac{\pi}{2}}^{\frac{\pi}{2}}\left(\frac{\partial\Psi_n}{\partial\beta}\cdot\frac{\partial(\Delta\Psi_n^2)}{\partial\alpha}-\frac{\partial\Psi_n}{\partial\alpha}\cdot\frac{\partial(\Delta\Psi_n^2)}{\partial\beta}\right)d\alpha d\beta$$
$$\leq\mu\int_0^{2\pi}\int_{-\frac{\pi}{2}}^{\frac{\pi}{2}}\left[\left(\frac{\partial\Psi_n}{\partial\beta}\right)^2+\Delta\Psi_n^2\right]\cos\alpha d\alpha d\beta \qquad (21)$$
$$=\mu\left(\left\|\frac{\partial\Psi_n}{\partial\beta}\right\|^2+\left\|\Delta\Psi_n\right\|^2\right).$$

As with Lemma 1 and Equation (21), it is not hard to verify the following:

$$\frac{\partial}{\partial t}\left\|\Delta\Psi_n\right\|^2\leq M_1\left\|\Delta\Psi_n\right\|^2. \qquad (22)$$

By integrating from 0 to $t$ on both sides of Equation (22) and applying the Gronwall inequality, we obtain

$$\left\|\Delta\Psi_n\right\|^2\leq\left\|\Delta\Psi_{0n}\right\|^2 e^{M_1 t}\leq\left(\left\|\Delta\Psi_{0n}-\Delta\Psi_0\right\|^2+\left\|\Delta\Psi_0\right\|^2\right)e^{M_1 t}\leq\widehat{C}_1. \qquad (23)$$

By multiplying both sides of Equations (17) and (18) by $\overline{C}_{lm}^{n}(t)$ and $\widetilde{C}_{lm}^{n}(t)$, respectively, and adding them in turn, we have

$$\int_0^{2\pi}\int_{-\frac{\pi}{2}}^{\frac{\pi}{2}}\left[\frac{\partial\Delta\Psi_n}{\partial t}+\frac{1}{\cos\alpha}\left(\frac{\partial\Psi_n}{\partial\beta}\frac{\partial\Delta\Psi_n}{\partial\alpha}-\frac{\partial\Psi_n}{\partial\alpha}\frac{\partial\Delta\Psi_n}{\partial\beta}\right)+2\mu\frac{\partial\Psi_n}{\partial\beta}\right]\Psi_n\cos\alpha d\alpha d\beta=0.$$

Hence, we have

$$\left|\int_0^{2\pi}\int_{-\frac{\pi}{2}}^{\frac{\pi}{2}}\left[\frac{\partial}{\partial t}\left(\frac{\partial}{\partial\alpha}\left(\cos\alpha\frac{\partial\Psi_n}{\partial\alpha}\right)+\frac{1}{\cos\alpha}\frac{\partial^2\Psi_n}{\partial\beta^2}\right)\right]\Psi_n\cos\alpha d\alpha d\beta\right.$$
$$\left.+\frac{1}{2}\int_0^{2\pi}\int_{-\frac{\pi}{2}}^{\frac{\pi}{2}}\left(\frac{\partial\Psi_n^2}{\partial\beta}\cdot\frac{\partial\Delta\Psi_n}{\partial\alpha}-\frac{\partial\Psi_n^2}{\partial\alpha}\cdot\frac{\partial\Delta\Psi_n}{\partial\beta}\right)d\alpha d\beta\right| \qquad (24)$$
$$\leq\mu\int_0^{2\pi}\int_{-\frac{\pi}{2}}^{\frac{\pi}{2}}\left[\left(\frac{\partial\Psi_n}{\partial\beta}\right)^2+\Psi_n^2\right]\cos\alpha d\alpha d\beta,$$

By applying the integration by parts formula, we have

$$\frac{\partial}{\partial t}\left(\left\|\frac{\partial\Psi_n}{\partial\alpha}\right\|^2+\left\|\frac{1}{\cos\alpha}\frac{\partial\Psi_n}{\partial\beta}\right\|^2\right)\leq 2\mu\left(\left\|\Psi_n\right\|^2+\left\|\frac{\partial\Psi_n}{\partial\beta}\right\|^2\right)$$
$$\leq 2\mu\left(\frac{1}{\lambda_0^2}\left\|\Delta\Psi_n\right\|^2+\left\|\frac{\partial\Psi_n}{\partial\alpha}\right\|^2+\left\|\frac{1}{\cos\alpha}\frac{\partial\Psi_n}{\partial\beta}\right\|^2\right) \qquad (25)$$
$$\leq M_2+2\mu\left(\left\|\frac{\partial\Psi_n}{\partial\alpha}\right\|^2+\left\|\frac{1}{\cos\alpha}\frac{\partial\Psi_n}{\partial\beta}\right\|^2\right).$$

When integrating from 0 to $t$ at both sides of Equation (25) and applying the Gronwall inequality again, there exists a $\widehat{C}_2 > 0$ such that

$$
\left\| \frac{\partial \Psi_n}{\partial \alpha} \right\|^2 + \left\| \frac{1}{\cos \alpha} \frac{\partial \Psi_n}{\partial \beta} \right\|^2 \leq \left[ M_2 T + \left\| \frac{\partial \Psi_0}{\partial \alpha} \right\|^2 + \left\| \frac{1}{\cos \alpha} \frac{\partial \Psi_0}{\partial \beta} \right\|^2 + \left\| \frac{\partial \Psi_{0n}}{\partial \alpha} - \frac{\partial \Psi_0}{\partial \alpha} \right\|^2 \right.
$$
$$
\left. + \left\| \frac{1}{\cos \alpha} \frac{\partial \Psi_{0n}}{\partial \beta} - \frac{1}{\cos \alpha} \frac{\partial \Psi_0}{\partial \beta} \right\|^2 \right] e^{2\mu t} \leq \widehat{C}_2.
$$

**Theorem 1.** *Assume that $(H)$ holds. Then, Equation (7) with the initial condition in Equation (8) and the boundary condition in Equation (9) has at least a weak solution.*

**Proof.** It is easy to verify that $\{\Delta \Psi_n\}$, $\left\{ \frac{\partial \Psi_n}{\partial \alpha} \right\}$, and $\left\{ \frac{1}{\cos \alpha} \frac{\partial \Psi_n}{\partial \beta} \right\}$ are uniformly weakly compact with respect to $t \in [0, T]$ in $L^2\left( (0, 2\pi) \times (-\frac{\pi}{2}, \frac{\pi}{2}) \right)$.

We are now going to show that $\left\{ \cos \alpha \frac{\partial \Psi_n}{\partial \alpha} \right\}$ is strongly compact with respect to $t \in [0, T]$ in $L^2\left( (0, 2\pi) \times (-\frac{\pi}{2}, \frac{\pi}{2}) \right)$. In fact, it is easy to know from the above statement that $\left\{ \cos \alpha \frac{\partial \Psi_n}{\partial \alpha} \right\}$ is uniformly bounded with respect to $t \in [0, T]$.

Let us show that $\left\{ \cos \alpha \frac{\partial \Psi_n}{\partial \alpha} \right\}$ is equicontinuous in $L^2\left( (0, 2\pi) \times (-\frac{\pi}{2}, \frac{\pi}{2}) \right)$. In fact, by using the mean value theorem and combining Equations (20) and (23), we obtain

$$
\left| \int_0^{2\pi} \int_{-\frac{\pi}{2}}^{\frac{\pi}{2}} \left( \cos(\alpha + \Delta\alpha) \frac{\partial \Psi_n}{\partial \alpha} (\alpha + \Delta\alpha, \beta + \Delta\beta, t) - \cos \alpha \frac{\partial \Psi_n}{\partial \alpha} (\alpha, \beta, t) \right)^2 \cos \alpha \, d\alpha \, d\beta \right|
$$
$$
= \left| \int_0^{2\pi} \int_{-\frac{\pi}{2}}^{\frac{\pi}{2}} \left[ \left( \frac{\partial}{\partial \alpha} \left( \cos(\alpha + \xi_1 \Delta\alpha) \frac{\partial \Psi_n}{\partial \alpha} (\alpha + \xi_1 \Delta\alpha, \beta + \Delta\beta, t) \right) \right) \cdot \Delta\alpha \right. \right.
$$
$$
\left. \left. + \left( \cos \alpha \frac{\partial^2 \Psi_n}{\partial \alpha \partial \beta} (\alpha, \beta + \xi_2 \Delta\beta, t) \right) \cdot \Delta\beta \right]^2 \cos \alpha \, d\alpha \, d\beta \right|
$$
$$
\leq 2 \left( \left\| \frac{\partial}{\partial \alpha} \left( \cos \alpha \frac{\partial \Psi_n}{\partial \alpha} \right) \right\|^2 \cdot (\Delta\alpha)^2 + \left\| \cos \alpha \frac{\partial^2 \Psi_n}{\partial \alpha \partial \beta} \right\|^2 \cdot (\Delta\beta)^2 \right)
$$
$$
\leq 2 \|\Delta \Psi_n\|^2 \left( (\Delta\alpha)^2 + (\Delta\beta)^2 \right)
$$
$$
\leq 2\widehat{C}_1 \left( (\Delta\alpha)^2 + (\Delta\beta)^2 \right),
$$

where $0 < \xi_1 < \xi_2 < 1$, which shows that $\left\{ \cos \alpha \frac{\partial \Psi_n}{\partial \alpha} \right\}$ is strongly compact in $L^2\left( (0, 2\pi) \times (-\frac{\pi}{2}, \frac{\pi}{2}) \right)$. Analogously, $\left\{ \frac{\partial \Psi_n}{\partial \beta} \right\}$ is strongly compact in $L^2\left( (0, 2\pi) \times (-\frac{\pi}{2}, \frac{\pi}{2}) \right)$.

Assume that

$$
\Psi_n \rightharpoonup \Psi \quad \text{weakly in } L^2\left( (0, 2\pi) \times (-\frac{\pi}{2}, \frac{\pi}{2}) \right) \quad \text{as } n \to \infty, \tag{26}
$$

Then, we have

$$
\frac{\partial \Psi_n}{\partial \alpha} \rightharpoonup \frac{\partial \Psi}{\partial \alpha} \qquad \text{weakly in } L^2\left( (0, 2\pi) \times (-\frac{\pi}{2}, \frac{\pi}{2}) \right) \quad \text{as } n \to \infty,
$$
$$
\frac{1}{\cos \alpha} \frac{\partial \Psi_n}{\partial \beta} \rightharpoonup \frac{1}{\cos \alpha} \frac{\partial \Psi}{\partial \beta} \quad \text{weakly in } L^2\left( (0, 2\pi) \times (-\frac{\pi}{2}, \frac{\pi}{2}) \right) \quad \text{as } n \to \infty,
$$
$$
\Delta \Psi_n \rightharpoonup \Delta \Psi \qquad \text{weakly in } L^2\left( (0, 2\pi) \times (-\frac{\pi}{2}, \frac{\pi}{2}) \right) \quad \text{as } n \to \infty.
$$

When setting $\frac{\partial \Psi_n}{\partial \alpha} \rightharpoonup \zeta_1(\alpha, \beta, t)$, $\frac{1}{\cos \alpha} \frac{\partial \Psi_n}{\partial \beta} \rightharpoonup \zeta_2(\alpha, \beta, t)$, and $\Delta \Psi_n \rightharpoonup \zeta_3(\alpha, \beta, t)$, for $\forall \ \xi(\alpha, \beta, t) \in C^1\left(\left(-\frac{\pi}{2}, \frac{\pi}{2}\right) \times (0, 2\pi) \times [0, T]\right)$, by the definition of the generalized derivative and Equation (26), we have

$$\int_0^{2\pi} \int_{-\frac{\pi}{2}}^{\frac{\pi}{2}} \zeta_1 \cdot \xi \cdot \cos \alpha d\alpha d\beta$$

$$= \lim_{n \to \infty} \int_0^{2\pi} \int_{-\frac{\pi}{2}}^{\frac{\pi}{2}} \frac{\partial \Psi_n}{\partial \alpha} \cdot \xi \cdot \cos \alpha d\alpha d\beta$$

$$= -\lim_{n \to \infty} \int_0^{2\pi} \int_{-\frac{\pi}{2}}^{\frac{\pi}{2}} \Psi_n \frac{\partial}{\partial \alpha} (\xi \cos \alpha) d\alpha d\beta$$

$$= -\int_0^{2\pi} \int_{-\frac{\pi}{2}}^{\frac{\pi}{2}} \Psi \frac{\partial}{\partial \alpha} (\xi \cos \alpha) d\alpha d\beta$$

$$= \int_0^{2\pi} \int_{-\frac{\pi}{2}}^{\frac{\pi}{2}} \frac{\partial \Psi}{\partial \alpha} \cdot \xi \cdot \cos \alpha d\alpha d\beta$$

which implies that $\frac{\partial \Psi}{\partial \alpha} = \zeta_1(\alpha, \beta, t)$. Similarly, we can verify that $\frac{1}{\cos \alpha} \frac{\partial \Psi}{\partial \beta} = \zeta_2(\alpha, \beta, t)$. On the other hand, we have

$$\int_0^{2\pi} \int_{-\frac{\pi}{2}}^{\frac{\pi}{2}} \zeta_3 \cdot \xi \cdot \cos \alpha d\alpha d\beta = \lim_{n \to \infty} \int_0^{2\pi} \int_{-\frac{\pi}{2}}^{\frac{\pi}{2}} \Delta \Psi_n \cdot \xi \cdot \cos \alpha d\alpha d\beta$$

$$= \lim_{n \to \infty} \int_0^{2\pi} \int_{-\frac{\pi}{2}}^{\frac{\pi}{2}} \left[ \frac{1}{\cos \alpha} \frac{\partial}{\partial \alpha} \left( \cos \alpha \frac{\partial \Psi_n}{\partial \alpha} \right) + \frac{1}{\cos^2 \alpha} \frac{\partial^2 \Psi_n}{\partial \beta^2} \right] \xi \cos \alpha d\alpha d\beta$$

$$= -\lim_{n \to \infty} \int_0^{2\pi} \int_{-\frac{\pi}{2}}^{\frac{\pi}{2}} \left[ \cos \alpha \frac{\partial \Psi_n}{\partial \alpha} \frac{\partial \xi}{\partial \alpha} + \frac{1}{\sin \alpha} \frac{\partial \Psi_n}{\partial \beta} \frac{\partial \xi}{\partial \beta} \right] d\alpha d\beta$$

$$= -\int_0^{2\pi} \int_{-\frac{\pi}{2}}^{\frac{\pi}{2}} \left[ \cos \alpha \frac{\partial \Psi}{\partial \alpha} \frac{\partial \xi}{\partial \alpha} + \frac{1}{\sin \alpha} \frac{\partial \Psi}{\partial \beta} \frac{\partial \xi}{\partial \beta} \right] d\alpha d\beta$$

$$= \int_0^{2\pi} \int_{-\frac{\pi}{2}}^{\frac{\pi}{2}} \Delta \Psi \cdot \xi \cdot \cos \alpha d\alpha d\beta,$$

which shows that $\Delta \Psi = \zeta_3$.

We prove that the following conclusion is true:

$$\lim_{n \to \infty} \int_0^{2\pi} \int_{-\frac{\pi}{2}}^{\frac{\pi}{2}} \left[ \frac{1}{\cos \alpha} \left( \frac{\partial \Psi_n}{\partial \beta} \frac{\partial \widetilde{\xi}}{\partial \alpha} \Delta \Psi_n - \frac{\partial \Psi_n}{\partial \alpha} \frac{\partial \widetilde{\xi}}{\partial \beta} \Delta \Psi_n \right) \right] \cos \alpha d\alpha d\beta$$

$$= \int_0^{2\pi} \int_{-\frac{\pi}{2}}^{\frac{\pi}{2}} \left[ \frac{1}{\cos \alpha} \left( \frac{\partial \Psi}{\partial \beta} \frac{\partial \widetilde{\xi}}{\partial \alpha} \Delta \Psi - \frac{\partial \Psi}{\partial \alpha} \frac{\partial \widetilde{\xi}}{\partial \beta} \Delta \Psi \right) \right] \cos \alpha d\alpha d\beta$$

where $\widetilde{\xi}(\alpha, \beta, t) = \cos^2 \alpha \ \overline{\xi}(\alpha, \beta, t), \overline{\xi}(\alpha, \beta, t) \in C^1\left(\left(-\frac{\pi}{2}, \frac{\pi}{2}\right) \times (0, 2\pi) \times [0, T]\right)$.

In fact, by simple calculation, we have

$$\left| \int_0^{2\pi} \int_{-\frac{\pi}{2}}^{\frac{\pi}{2}} \left[ \frac{1}{\cos \alpha} \left( \frac{\partial \Psi_n}{\partial \beta} \Delta \Psi_n \left( -2 \cos \alpha \sin \alpha \overline{\xi} + \cos^2 \alpha \frac{\partial \overline{\xi}}{\partial \alpha} \right) \right. \right. \right.$$

$$- \frac{\partial \Psi}{\partial \beta} \Delta \Psi \left( -2 \cos \alpha \sin \alpha \overline{\xi} + \cos^2 \alpha \frac{\partial \overline{\xi}}{\partial \alpha} \right) - \frac{\partial \Psi_n}{\partial \alpha} \Delta \Psi_n \frac{\partial \overline{\xi}}{\partial \beta} \cos^2 \alpha$$

$$\left. \left. \left. + \frac{\partial \Psi}{\partial \alpha} \Delta \Psi \frac{\partial \overline{\xi}}{\partial \beta} \cos^2 \alpha \right) \right] \cos \alpha d\alpha d\beta \right|$$

$$\leq \left| \int_0^{2\pi} \int_{-\frac{\pi}{2}}^{\frac{\pi}{2}} \left[ \left( \cos \alpha \frac{\partial \Psi_n}{\partial \alpha} - \cos \alpha \frac{\partial \Psi}{\partial \alpha} \right) \Delta \Psi_n \frac{\partial \overline{\xi}}{\partial \beta} \cos \alpha \right] d\alpha d\beta \right|$$

$$+\left|\int_0^{2\pi}\int_{-\frac{\pi}{2}}^{\frac{\pi}{2}}\left[\left(\cos\alpha\frac{\partial\Psi}{\partial\alpha}(\Delta\Psi_n-\Delta\Psi)\frac{\partial\overline{\widetilde{\zeta}}}{\partial\beta}\right)\cos\alpha\right]d\alpha d\beta\right|$$

$$+\left|\int_0^{2\pi}\int_{-\frac{\pi}{2}}^{\frac{\pi}{2}}\left[\left(\left(\frac{\partial\Psi_n}{\partial\beta}-\frac{\partial\Psi}{\partial\beta}\right)\cdot\left(\frac{\partial\overline{\widetilde{\zeta}}}{\partial\alpha}\cos\alpha-2\overline{\widetilde{\zeta}}\sin\alpha\right)\right)\Delta\Psi_n\cos\alpha\right]d\alpha d\beta\right|$$

$$+\left|\int_0^{2\pi}\int_{-\frac{\pi}{2}}^{\frac{\pi}{2}}\left[\left(\left(\frac{\partial\overline{\widetilde{\zeta}}}{\partial\alpha}\cos\alpha-2\overline{\widetilde{\zeta}}\sin\alpha\right)\cdot(\Delta\Psi_n-\Delta\Psi)\right)\frac{\partial\Psi}{\partial\beta}\cos\alpha\right]d\alpha d\beta\right|$$

$$<\varepsilon.$$

Suppose $\widetilde{\zeta}(\alpha,\beta,t)$ has the following approximate solution:

$$\widetilde{\zeta}_n(\alpha,\beta,t)=\cos^2\alpha\,\overline{\widetilde{\zeta}}_n(\alpha,\beta,t)=\sum_{l=0}^n\sum_{m=0}^l(\overline{c}_{lm}(t)\cos m\beta+\widetilde{c}_{lm}(t)\sin m\beta)P_l^m(\sin\alpha),$$

where

$$\overline{c}_{lm}(t)=\frac{1}{\Pi_{lm}}\int_0^{2\pi}\int_{-\frac{\pi}{2}}^{\frac{\pi}{2}}\left(\cos^2\alpha\,\overline{\widetilde{\zeta}}(\alpha,\beta,t)\cos m\beta P_l^m(\sin\alpha)\cos\alpha\right)d\alpha d\beta,$$

$$\widetilde{c}_{lm}(t)=\frac{1}{\Pi_{lm}}\int_0^{2\pi}\int_{-\frac{\pi}{2}}^{\frac{\pi}{2}}\left(\cos^2\alpha\,\overline{\widetilde{\zeta}}(\alpha,\beta,t)\sin m\beta P_l^m(\sin\alpha)\cos\alpha\right)d\alpha d\beta.$$

By multiplying Equations (17) and (18) by $\overline{c}_{lm}(t)$ and $\widetilde{c}_{lm}(t)$, respectively, and then adding them together and integrating from 0 to $t$, we have

$$\int_0^t\int_0^{2\pi}\int_{-\frac{\pi}{2}}^{\frac{\pi}{2}}\left[\left(\frac{\partial\Delta\Psi_n}{\partial t}+\frac{1}{\cos\alpha}\left(\frac{\partial\Psi_n}{\partial\beta}\frac{\partial\Delta\Psi_n}{\partial\alpha}-\frac{\partial\Psi_n}{\partial\alpha}\frac{\partial\Delta\Psi_n}{\partial\beta}\right)+2\mu\frac{\partial\Psi_n}{\partial\beta}\right)\widetilde{\zeta}_n\cos\alpha\right]d\alpha d\beta dt=0,$$

Then, by using the integration by parts formula, we obtain

$$\int_0^t\int_0^{2\pi}\int_{-\frac{\pi}{2}}^{\frac{\pi}{2}}\left[\left((-\Delta\Psi_n)\frac{\partial\widetilde{\zeta}_n}{\partial t}-\frac{1}{\cos\alpha}\left(\frac{\partial\Psi_n}{\partial\beta}\frac{\partial\widetilde{\zeta}_n}{\partial\alpha}-\frac{\partial\Psi_n}{\partial\alpha}\frac{\partial\widetilde{\zeta}_n}{\partial\beta}\right)\Delta\Psi_n\right.\right.$$

$$\left.\left.+2\mu\widetilde{\zeta}_n\frac{\partial\Psi_n}{\partial\beta}\right)\cos\alpha\right]d\alpha d\beta dt+\int_0^{2\pi}\int_{-\frac{\pi}{2}}^{\frac{\pi}{2}}\Delta\Psi_{0n}(\alpha,\beta)\cos\alpha d\alpha d\beta=0. \tag{27}$$

Now let $n\to\infty$ in Equation (27). Then, the conclusion of Theorem 1 is proven.　□

Next, we give the following non-uniqueness of weak solutions:

**Theorem 2.** *Suppose* $\Psi(\alpha,\beta,t)$ *is a weak solution to Equation (7) with the initial condition in Equation (8) and the boundary condition in Equation (9). For any function where* $\widetilde{\Psi}(t)\in C^1[0,T]$ *and* $\widetilde{\Psi}(0)=0$ *, then* $\overline{\Psi}(\alpha,\beta,t)=\Psi(\alpha,\beta,t)+\widetilde{\Psi}(t)$ *is also a weak solution to Equation (7) with the initial condition of Equation (8) and boundary condition of Equation (9).*

**Proof.** Noting that $\Psi$ is a weak solution to Equation (7), Equation (27) holds. Based on the assumption of the Theorem 2, we obtain

$$\int_0^t\int_0^{2\pi}\int_{-\frac{\pi}{2}}^{\frac{\pi}{2}}\left[\left((-\Delta\overline{\Psi})\frac{\partial\widetilde{\zeta}}{\partial t}-\frac{1}{\cos\alpha}\left(\frac{\partial\overline{\Psi}}{\partial\beta}\frac{\partial\widetilde{\zeta}}{\partial\alpha}-\frac{\partial\overline{\Psi}}{\partial\alpha}\frac{\partial\widetilde{\zeta}}{\partial\beta}\right)\Delta\overline{\Psi}\right.\right.$$

$$\left.\left.+2\mu\widetilde{\zeta}\frac{\partial\overline{\Psi}}{\partial\beta}\right)\cos\alpha\right]d\alpha d\beta dt+\int_0^{2\pi}\int_{-\frac{\pi}{2}}^{\frac{\pi}{2}}\Delta\overline{\Psi}_0(\alpha,\beta)\cos\alpha d\alpha d\beta=0,$$

which implies that Theorem 2 is correct.　□

## 4. Conclusions

According to the concept of spherical harmonic function and weak solutions, the governing equation describing atmospheric flow is transformed into a second-order homogeneous linear ordinary differential equation and a Legendre's differential equation by using the method of separating variables. In this paper, the existence and non-uniqueness of weak solutions of vorticity equations are obtained by using the Gronwall inequality and Cauchy–Schwartz inequality under the appropriate initial and boundary conditions. Compared with the vorticity equation established in [5], our conclusion considers the time dependence of the system. The authors of [19] studied the stability of the Rossby–Haurwitz stationary solution but did not provide the existence proof for the solution. In this paper, we give the existence results of weak solutions, which to a certain extent provides a theoretical basis for the stability study of [19].

**Author Contributions:** The contributions of all authors are equal. All authors have read and agreed to the published version of the manuscript.

**Funding:** This work is partially supported by Guizhou Provincial Science and Technology Projects (ZK[2022]535), Training Object of High Level and Innovative Talents of Guizhou Province ((2016)4006), Major Research Project of Innovative Group in Guizhou Education Department ([2018]012), the Slovak Research and Development Agency under the contract No. APVV-18-0308, and by the Slovak Grant Agency VEGA No. 1/0358/20 and No. 2/0127/20.

**Institutional Review Board Statement:** Not applicable.

**Informed Consent Statement:** Not applicable.

**Data Availability Statement:** Not applicable.

**Conflicts of Interest:** The authors declare no conflict of interest.

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
