# Peer review of "The Existence of Weak Solutions for the Vorticity Equation Related to the Stratosphere in a Rotating Spherical Coordinate System"

_axioms, doi:10.3390/axioms11070347_

Round 1
Reviewer 1 Report
The authors of the paper consider the time-dependent stratospheric planetary flow in a rotating spherical coordinate system. Pursuant to the concepts of spherical harmonic functions and weak solutions, they have obtained the existence and non-uniqueness of weak solutions for nonlinear governing equations under the suitable boundary conditions. The stability of the critical stationary solution is studied by the authors in the limit case in which the stream function belongs to the sum of the first two eigenspaces of Laplace-Beltrami operator. Local and global bifurcation results for nonzonal stationary solutions of classical Rossby-Haurwitz waves have been provided as well. The paper is structured and presented in appropriate way. The mathematical steps related to the theorems have been explained and presented in a correct way and in an original manner. The academic terminology is also correctly used.
Please find below my humble suggestions to improve the paper:
The novelty of the paper can be stated in a more clear way, if possible.
By comparing the study with the previous ones, the contributions of the paper can be expressed as well.
The solutions can be supported by the related examples in line with the proven theorems.
Following these above mentioned minor revision aspects, the paper can be published.
Yours faithfully,
Author Response
1. The novelty of the paper can be stated in a more clear way, if possible.
\textbf{Answer}: Done.
2. By comparing the study with the previous ones, the contributions of the paper can be expressed as well.
\textbf{Answer}: Done.
3. The solutions can be supported by the related examples in line with the proven theorems
\textbf{Answer}: Thank you. Existence of weak solutions of vorticity equation is proved strictly according to the definition of weak solutions in partial differential equation.
Reviewer 2 Report
The abstract is very superficial : no context, no any results, no any benefits – pleas reframe it in a proper manner
What is “MSC: 35Q35”
Please avoid block citation such as [8-16]; maximum 3 it will be enough. And it is requested to discuss them individually
From let say method section “Stratospheric planetary flows equation” it is not clear which is author contribution as long everything was endorsed by different citations !
A case study to exemplify the assumption proposed is required
There is need a section for discussion
The conclusion are missing
Author Response
1. The abstract is very superficial: no context, no any results, no any benefits-pleas reframe it in a proper manner.
\textbf{Answer}: Thank you. Corrected.
2. What is ``MSC: 35Q35" ?
\textbf{Answer}: ``MSC" is short for ``the Mathematics Subject Classification". ``35Q35" represents ``PDEs in connection with fluid mechanics".
3. Please avoid block citation such as [8-16]; maximum 3 it will be enough. And it is requested to discuss them individually.
\textbf{Answer}: Thank you. Corrected.
4. From let say method section ``Stratospheric planetary flows equation" it is not clear which is author contribution as long everything was endorsed by different citations !
\textbf{Answer}: Our main contribution is to transform the governing equation describing atmospheric flow into a second order homogeneous linear ordinary differential equation and a Legendre's differential equation by using the method of separating variables according to the concepts of spherical harmonic functions and weak solutions. Existence and non-uniqueness of weak solutions of vorticity equations are obtained by using the Gronwall inequality and Cauchy-Schwartz inequality under appropriate initial and boundary conditions.
5. A case study to exemplify the assumption proposed is required.
\textbf{Answer}: Thank you. It is not difficult to see that compactness cannot be guaranteed without the assumption. We can see the proof of Theorem 3.
6. There is need a section for discussion.
\textbf{Answer}: Done.
7. The conclusion are missing.
\textbf{Answer}: Thank you. We have added the conclusion.
Reviewer 3 Report
The paper is very interesting and can be published after the authors will address the following major comments:
- The abstract should be extended in order to include the main results of the paper and the method that the authors used to achieve these results.
- Nomenclature must be added to the paper.
- Every variable and parameter must be explained in the paper and the author should write the physical unit of every parameter and variable
- The introduction section must be extended in order to include more literature that is relevant to the current study.
- The transition from the dimensional model to the dimensionless model must be explained in detail.
- The author applies a partial derivative after the Laplace operator. This should be explained (see eq. 7)
- Eq. 17, 18, 20......... the integral didn't define at pi/2 and -pi/2 since at the denominator there is cos.
- What norm did the authors use in Lemma2?
- Conclusion section must be added to the paper.
Author Response
1. The abstract should be extended in order to include the main results of the paper and the method that the authors used to achieve these results.
\textbf{Answer}: Thank you. Corrected.
2. Nomenclature must be added to the paper.
\textbf{Answer}: Thank you. Added.
3. Every variable and parameter must be explained in the paper and the author should write the physical unit of every parameter and variable.
\textbf{Answer}: Thank you. See table 1 on page 3.
4. The introduction section must be extended in order to include more literature that is relevant to the current study.
\textbf{Answer}: We added more major works to the literature related to the current study.
5. The transition from the dimensional model to the dimensionless model must be explained in detail.
\textbf{Answer}: Thank you. In the third page, the transformation process from dimensional model to dimensionless model is given in detail.
6. The author applies a partial derivative after the Laplace operator. This should be explained (see eq. 7).
\textbf{Answer}: Equation (7) is obtained by substituting (6) into (4).
7. Eq. 17, 18, 20 ...... the integral didn't define at pi/2 and -pi/2 since at the denominator there is cos.
\textbf{Answer}: Thank you. We have multiplied each term of the Eq. 17, 18, 20 ...... by a factor ``cos", which we have taken into account in the definition of the norm.
8. What norm did the authors use in Lemma 2 ?
\textbf{Answer}: Define the norm as follows
\begin{align*}
\|\psi\|^{2} = \int_{0}^{2\pi}\int_{-\frac{\pi}{2}}^{\frac{\pi}{2}} \psi^{2} \cos \alpha d\alpha d\beta ~~\text{for}~~ \psi \in L^{2}\left((0,2\pi) \times (-\frac{\pi}{2},\frac{\pi}{2})\right).
\end{align*}
9. Conclusion section must be added to the paper.
\textbf{Answer}: Thank you. We have added the conclusion.
Round 2
Reviewer 2 Report
-
Author Response
Thank you. Corrected.
Reviewer 3 Report
I read again the paper and it seems that the author didn't address my major comments. For example:
1: Nomenclature must be added to the paper.
2: Every variable and parameter must be explained in the paper and the author should write the physical unit of every parameter and variable
3: The transition from the dimensional model to the dimensionless model must be explained in detail.
4: The author applies a partial derivative after the Laplace operator. This should be explained (see eq. 7)
5: Eq. 17, 18, 20......... the integral didn't define at pi/2 and -pi/2 since at the denominator there is cos.
6: What norm did the authors use in Lemma2?
7: The conclusion section must be added to the paper.
Round 3
Reviewer 3 Report
The authors revised the paper according to my comments except from comment number 7.
Author Response
- Conclusion section must be added to the paper.
Answer: Thank you. We added the conclusion section on page 13 of the paper in blue
font.